# Vitamin C-Assisted Fabrication of Aerogels from Industrial Graphene Oxide for Gaseous Hexamethyldisiloxane Adsorption

Yanhui Zheng [1,2], Xifeng Hou [1], Xiaolong Ma [3,*], Zelin Hao [1] and Zichuan Ma [1,*]

1 Hebei Key Laboratory of Inorganic Nano-Materials, College of Chemistry and Material Science, Hebei Normal University, Shijiazhuang 050024, China; zhengyh0308@163.com (Y.Z.); xifenghoucc@163.com (X.H.); haozelin1999@163.com (Z.H.)
2 Shijiazhuang Vocational College of Finance & Economics, Shijiazhuang 050061, China
3 School of Environmental Science and Engineering, Hebei University of Science and Technology, Shijiazhuang 050018, China
* Correspondence: maxiaolong2410@hebust.edu.cn (X.M.); mazc@hebtu.edu.cn (Z.M.); Tel.: +86-311-80787400 (Z.M.)

**Abstract:** Volatile methyl siloxanes (VMSs) as a trace impurity in biogas decreases its energy utilization, and thus need to be removed. In this paper, a one-step hydrothermal reduction was performed to produce three-dimensional reduced graphene oxide aerogels (rGOAs) using industrial-grade graphene oxide (IGGO) as raw material and vitamin C (VC) as a reductant to facilitate the fabrication of rGOAs. The synthesis of rGOAs was a simple, green, and energy-efficient process. The developed rGOAs were characterized using the Brunauer–Emmett–Teller method, Raman spectrometry, scanning electron microscopy, Fourier-transform infrared spectroscopy, X-ray diffraction measurements and contact angle. The results obtained showed that rGOA-1 with a VC/IGGO ratio of 1/1 (m/m) exhibited a hierarchical porous structure and super-hydrophobicity, yielding a high specific surface area (137.9 m$^2$ g$^{-1}$) and superior water contact angle (143.8°). The breakthrough adsorption capacity of rGOA-1 for hexamethyldisiloxane (L2, a VMS model) was 11 times higher than that of IGGO. Low inlet concentration and bed temperature were considered beneficial for the L2 adsorption. Interestingly, rGOA-1 was less sensitive to water, and it was readily regenerated for reuse by annealing at 80 °C. The rGOAs have been demonstrated to have great potential for the removal of siloxanes from biogas.

**Keywords:** reduced graphene oxide aerogel; Vitamin C; hexamethyldisiloxane; siloxane; adsorption

## 1. Introduction

Volatile methyl siloxanes (VMSs) are products of the hydrolytic depolymerization of silicone. In VMS molecules, silicon atoms are alternately linked to oxygen atoms and methyl groups, exhibiting cyclic and chain structures labeled D and L, respectively (Figure S1) [1–3]. Due to the widespread use of silicone, the VMS content in biogas originating from the digestion of organic waste substances in sewage plants, municipal landfills, and farm fermentation tanks has recently grown quickly [4–6]. Their presence in biogas is proven to impose significant constraints on its energy application, for example, blocking the fan and wearing out the fenestrae of the engine, etc., which are fatal to the biogas power equipment, substantially increasing the production cost [3]. Therefore, the VMS removal from biogas has recently come into research focus.

There are several available methods to remove VMSs, such as adsorption, absorption, biofiltration, and membrane filtering [7–9]. Among these techniques, adsorption has been the most widely used method because of higher efficiency, less risk, and lower cost than other approaches. Silica gel, activated carbons (ACs), and zeolites were applied in adsorbing VMSs, exhibiting good activity toward the VMS capture [10–15]. However, in the presence of moisture, the siloxane polymerization reaction occurs attributed to the

hydroxyl groups on the surface of ACs, zeolites, and silica gel, making thermal regeneration more difficult [16–19]. Thus, novel materials for the VMS adsorption need to be developed.

Graphene-based materials, for instance, graphene oxide (GO) and reduced graphene-oxide aerogels (rGOAs), have attracted significant attention for adsorbent applications [20–23]. Graphene oxide is comprised of many functional groups on its edges and basal planes, including epoxide, hydroxyl, carboxyl, and carbonyl groups [24–26]. The hydrothermal reduction can be used to eliminate the GO functional groups, yielding rGOAs [27]. Compared with GO, rGOAs show better performance of adsorption toward non-polar pollutants as a consequence of their high surface area, high hydrophobicity, and low oxygen content, suggesting great application potential of rGOAs for the VMS removal [28–31].

Up to now, a wide variety of reducing agents were studied for the preparation of rGOAs, such as $NH_3 \cdot H_2O$, HI, $Na_2S$, $C_6H_4(OH)_2$, $NaHSO_3$, Vitamin C (VC), etc. Among these reductants, VC was proved to be environmentally friendly [32–35]. In this study, we prepared rGOAs from industrial-grade multilayer graphene oxide (IGGO) using VC as a reductant. The reduction of IGGO was achieved at a low hydrothermal temperature. The prepared rGOAs demonstrated great potential for the VMS removal, easy recovery, and water resistivity. To the best of our knowledge, previous reports regarding the effects of VC on self-assembly of rGOAs in the hydrothermal reaction and their performance for the VMS removal are rather scarce; thus, the VC-assisted fabrication of aerogels has promising industrial applications.

## 2. Materials and Methods

### 2.1. Chemicals and Reagents

IGGO, industrial-grade, was purchased from Suzhou Hengqiu Technology Co., Ltd. (Suzhou, China). Vitamin C (99% purity, Aladdin, China), absolute ethanol (99.9% purity, Sairuifu Technology Co., Ltd., Tianjin, China), and deionized water were used for the synthesis of rGOAs. L2 (hexamethyldisiloxane), 99% purity, was obtained from Aladdin (Shanghai, China). All reagents were used directly without further purification.

### 2.2. Preparation of Reduced Graphene-Oxide Aerogels (rGOAs)

Aqueous suspensions of IGGO (4.0 mg $mL^{-1}$) were prepared with 240 mg of IGGO dispersing in 60 mL deionized water via ultrasonic treatment for 60 min. Briefly, 240 mg of VC was dissolved in an IGGO suspension, and the mixed suspension was transferred to a 100 mL Teflon reaction kettle, heated to 95 °C and kept 6 h, leading to the formation of hydrogel. Then, to remove the residual agents, the hydrogel was washed with deionized water, followed by immersing in 20% ethanol solution for 6 h. Finally, the rGOA-1 sample was achieved by freeze-drying for 24 h with an IGGO to VC mass ratio of 1:1. To explore the effect of VC on the rGOAs, we also prepared rGOAs with the other three IGGO to VC mass ratios of (1:0, 1:0.5 and 1:2), and the aerogel samples were denoted as rGOA-0, rGOA-0.5, and rGOA-2, respectively. Next, the rGOAs were ground and passed through a 100-mesh sieve for the following experiments to maintain consistency of the experimental conditions.

### 2.3. Characterization

The S-4800 scanning electron microscope (SEM, Hitachi, Japan) was used to examine the morphology of the materials. The X-ray diffraction (XRD, Bruker AXS, Karlsruhe, Germany) was used for the structural characterization, and Raman spectra were performed using a Raman spectrometer (Renishaw, NewMills, UK). The Kubo ×1000 automatic surface area and pore analyzer (Beijing Builder Co., Ltd., Beijing, China) were used to measure the $N_2$ adsorption-desorption isotherms at a temperature of −196 °C, with $10^{-5} < P/P_0 < 1.0$. The samples were treated under vacuum at 80 °C for 2 h. The BET-specific surface area ($S_{BET}$) was calculated by the Brunauer–Emmett–Teller (BET) equation. The total pore volume ($V_{tot}$) and the average pore size ($D_{aver}$) were calculated using the Barrett–Joyner–Halenda (BJH) method. The presence of functional groups on the samples were measured by the Fourier-transform infrared spectroscopy (FTIR, Bruker

Tensor 27, Ettlingen, Germany) in the range of 400–4000 cm$^{-1}$. The water contact angle was measured for a 3 µL water drop at room temperature using a contact angle/interface system (JY-PHb, Jinhe Instrument Manufacturing Co., LTD, Nanjing, Jiangsu, China) to determine the hydrophobicity of the samples. The L2 concentration was determined using GC 9790 gas chromatography (Fuli Analytical Instrument Co., Ltd., Wenling, Zhejiang, China) with a hydrogen flame ionization detector (GC/FID) and a 2.0 m long × 2.0 mm I.D. stationary-phase column (GDX-102). Other detection conditions are as follows: an oven temperature of 200 °C, an injector temperature of 250 °C, and a detector temperature of 280 °C, with an N$_2$ carrier gas flow rate of 50 mL min$^{-1}$.

### 2.4. L2 Adsorption and Regeneration

The dynamic adsorption experiments of L2 on rGOAs were performed according to the details reported in literature [36]. The experimental parameters, such as the mass of rGOAs, were set to 0.10 g at 20 °C and the L2 inlet concentration ($C_{in}$) of 14.62 mg L$^{-1}$ at a gas flow rate of 50 mL min$^{-1}$. $C_{out,t}$ (the L2 concentration in the gas line after adsorbing process) was detected via a calibrated GC-FID on the permeate side. The breakthrough curves by plotting $C_{out,t}/C_{in}$ versus time were used to express the experimental results. The following quantities were adopted to denote the adsorption behavior: (i) the breakthrough time ($t_B$, min), defined as $C_{out,t}/C_{in} = 0.05$, and (ii) the breakthrough adsorption capacity ($Q_B$, mg g$^{-1}$), representing the adsorption capacity at time $t_B$. The $Q_B$ was calculated from Equation (1) [36]:

$$Q_B = \frac{V_g C_{in}}{m} \int_0^{t_B} \left(1 - \frac{C_{out,t}}{C_{in}}\right) dt \qquad (1)$$

where $m$ is the mass of adsorbent (g), $V_g$ is the gas flow rate (L min$^{-1}$), $C_{in}$ is the inlet concentration (mg L$^{-1}$), and $C_{out,t}$ is the outlet concentration (mg L$^{-1}$) at adsorption time $t$ (min).

With $C_{out,t}/C_{in} \approx 1$, the adsorbent bed reached saturation. Next, the used rGOAs were regenerated with nitrogen blowing and heated in a water bath for 30 min at 80 °C. The five times adsorption-desorption cycles were repeated in the same manner to analyze the regenerative properties of rGOAs.

### 2.5. Model of the Breakthrough Curves

The Yoon–Nelson model was selected to assess the dynamic data of the experiment. The data obtained from this model can be used to calculate the theoretical breakthrough time ($t_{B,th}$, min) and theoretical breakthrough adsorption capacity ($Q_{B,th}$, mg g$^{-1}$). The Equation (2) for this model is [37]:

$$\frac{C_{out,t}}{C_{in}} = \frac{1}{1 + \exp[K_{YN}(\tau - t)]} \times 100\% \qquad (2)$$

where $K_{YN}$ is the constant of Yoon–Nelson, $\tau$ is the time by $C_{out,t}/C_{in} = 0.5$, and $t$ is the adsorption time (min).

The kinetic model parameters ($K_{YN}$ and $\tau$) of different adsorption experiments were obtained from the Equation (2). Then, for $C_{out,t}/C_{in} = 0.05$, $t_{B,th}$ was obtained by the Equation (2). After that, using the expressions of $C_{out,t}/C_{in}$ in the Equation (2), we can calculate the $Q_{B,th}$ by the Equation (1) at $t_{B,th}$.

## 3. Results

### 3.1. Effect of the Vitamin C (VC) Amount on Textural Properties and Hydrophobicity

Figure S2 shows digital photos of the rGOAs. The as-prepared samples exhibit different macroscopic appearances with and without VC. When VC is added, the rGOAs exist as a monolithic bulk with a stable cylindrical shape, as shown in Figure S2b–d. By contrast, the rGOA-0 yields only loose black mass powder (Figure S2a). It indicates that VC is beneficial to the fabrication of the rGOAs, yielding a three-dimensional monolithic

porous architecture of aerogel via π–π interactions, hydrophobicity, hydrogen bonds, and Van der Waals forces [20,31]. In addition, the amount of VC has a positive effect on the structure and functionality of the rGOAs. To assess the textural properties of IGGO and rGOAs, textural parameters ($S_{BET}$, $V_{tot}$, $D_{aver}$, and contact angle) are listed in Table 1. As evident from Table 1, the rGOAs exhibit a significant improvement in the textural properties, where rGOA-1 has the highest values of $S_{BET}$ and $V_{tot}$. For comparison, we chose the original samples of IGGO, rGOA-0 (without VC) and rGOA-1 (with highest specific surface area) to perform the rest of the experiments. Figure 1a,b) shows the nitrogen adsorption-desorption isotherms and the pore size distribution of IGGO and two representative rGOA composites (i.e., rGOA-0 and rGOA-1). As shown in Figure 1a, from $N_2$ adsorption–desorption isotherms, rGOA-1 presents a IUPAC type IV isotherm with a type H3 hysteresis loop [38], suggesting the existence of an external surface and mesopores [30]. On the other hand, IGGO and rGOA-0 have a much smaller nitrogen adsorption volume than rGOA-1, exhibiting type III isotherms, with nearly non-porous structure [21]. Remarkably, rGOA-1 exhibits narrower mesopores than IGGO and rGOA-0, as shown in Figure 1b. According to the literature [30], a narrower mesoporous range is favorable for the adsorption of L2.

**Table 1.** Properties of industrial-grade multilayer graphene oxide (IGGO) and reduced graphene-oxide aerogels (rGOAs).

| Adsorbent | $S_{BET}$ (m$^2$ g$^{-1}$) | $V_{tot}$ (cm$^3$ g$^{-1}$) | $D_{aver}$ (nm) | Contact Angle (°) |
|---|---|---|---|---|
| IGGO | 7.4 | 0.23 | 13.18 | 76.6 |
| rGOA-0 | 19.1 | 0.39 | 10.30 | 119.9 |
| rGOA-0.5 | 86.8 | 0.71 | 9.20 | 135.1 |
| rGOA-1 | 137.9 | 0.88 | 5.08 | 143.8 |
| rGOA-2 | 76.1 | 0.65 | 8.76 | 132.0 |

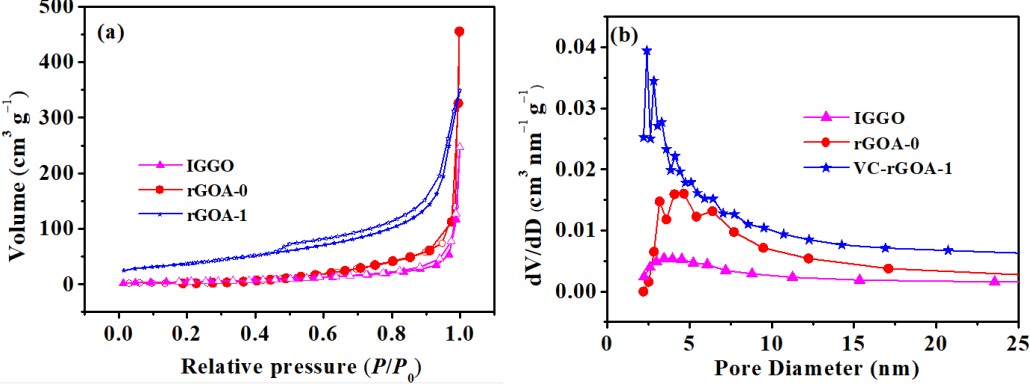

**Figure 1.** The nitrogen adsorption–desorption isotherms (**a**) and pore size distribution (**b**) by IGGO, rGOA-0, and rGOA-1.

The photos measuring contact angles are shown in Figure S3. The contact angle measurements of the adsorbents are listed in Table 1. The hydrophobicity of the rGOAs is stronger than that of IGGO, which might be explained by the removal of oxygen-containing groups of IGGO upon the reduction. In addition, rGOA-1 has the highest contact angle of 143.8°, showing super-hydrophobicity.

### 3.2. Scanning Electron Microscopy (SEM), X-ray Diffraction (XRD), Fourier-Transform Infrared (FTIR), and Raman Analyses of Industrial-Grade Graphene Oxide (IGGO), rGOA-0, and rGOA-1

SEM micrographs of IGGO, rGOA-0, and rGOA-1 are shown in Figure 2. For IGGO, they confirm a certain degree of aggregation of multilayered graphene sheets (Figure 2a,b). For rGOA-0, a few nanosheets are stacked and entangled, exhibiting no formation of a crosslinked structure (Figure 2c,d), perhaps due to insufficient hydrothermal self-assembly [30]. In particular, rGOA-1 shows a rich 3D porous network structure

(Figure 2e,f), and the pores seem to be formed with many wrinkles from the curling of sheets [39].

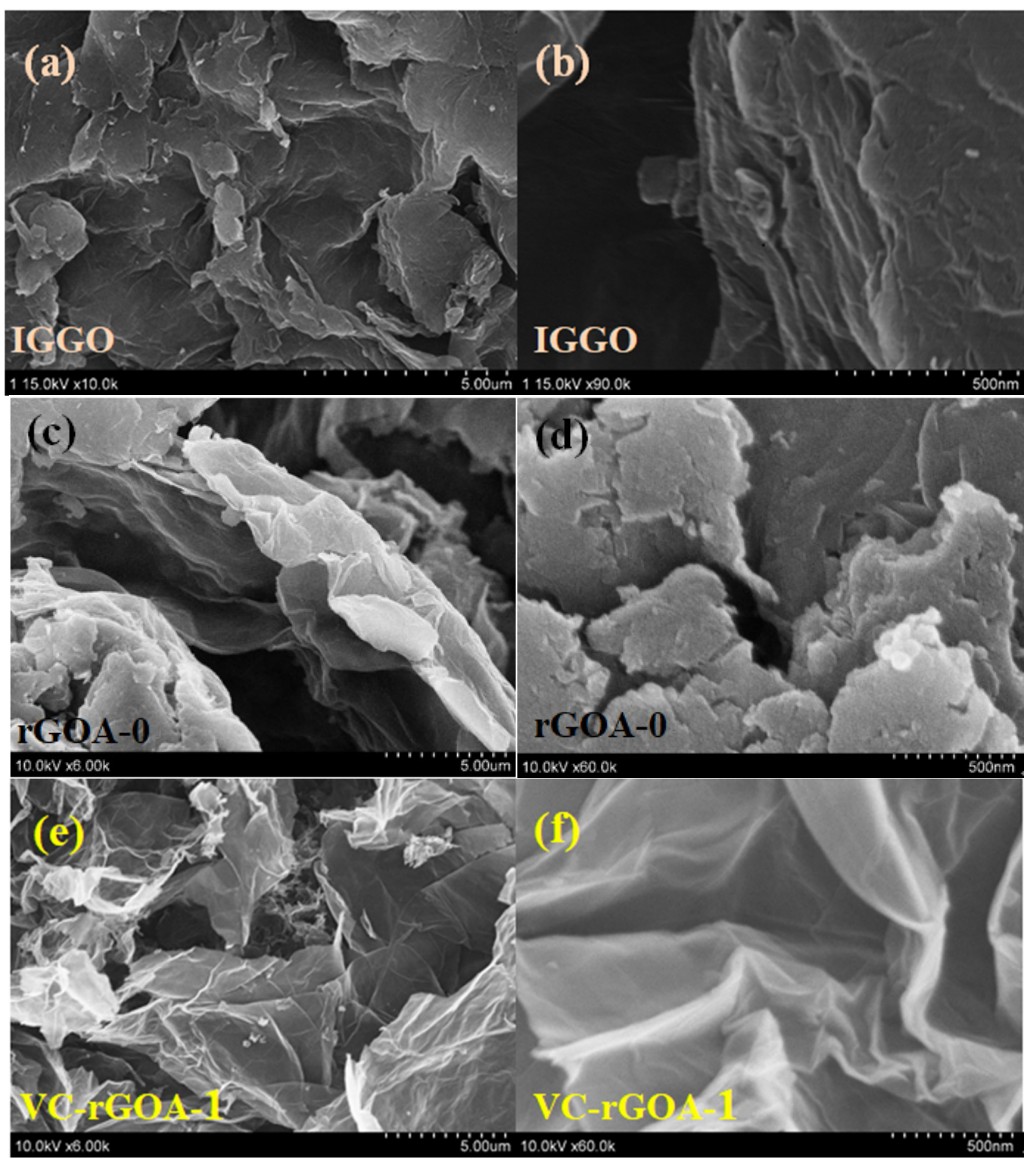

**Figure 2.** Scanning electron microscopy (SEM) images of IGGO (**a,b**), rGOA-0 (**c,d**), and rGOA-1 (**e,f**).

The structures of IGGO, rGOA-0, and rGOA-1 are further analyzed by XRD. The relevant XRD patterns (Figure 3) exhibit an significant structural change after the hydrothermal reduction. A typical, strong peak at $2\theta = 11.6°$, is the (001) plane of IGGO, with an interlayer spacing of 7.6 Å. This large distance of interlayer could be linked to the presence of epoxy, hydroxyl, and carboxyl groups [40]. The (001) peak of rGOA-0 decreases, and a faint (002) facet of graphitic carbon appears at $2\theta = 25.3°$ [41]. It proves that incomplete reduction occurs on the rGOA-0 nanosheets surface, which agrees with the SEM observations. After the reduction process, the XRD pattern of rGOA-1 has a (002) plane at $2\theta = 25.3°$, suggesting ordered graphene layers and yielding a smaller interlayer spacing of 3.6 Å. It could be claimed that due to the removal of the oxygen functionalities, C=C bonds and $\pi$–$\pi$ interactions are restored, reducing the interlayer spacing [42].

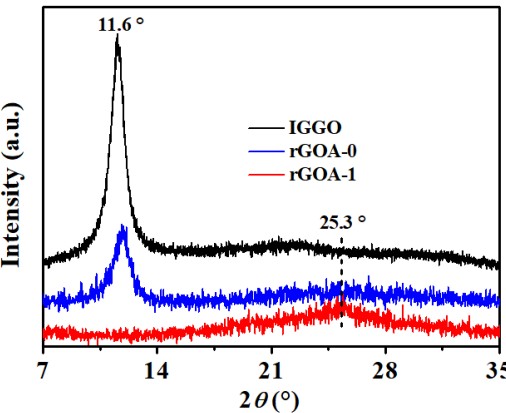

**Figure 3.** X-ray diffraction (XRD) patterns of IGGO, rGOA-0, and rGOA-1.

The Raman spectra of IGGO, rGOA-0, and rGOA-1 are given in Figure 4. Two characteristic bands, D band (1345 cm$^{-1}$) and G band (1600 cm$^{-1}$), represent the defects caused by sp$^3$-hybridized carbon atoms and the stretching vibrations by sp$^2$ lattice carbon atoms in the graphene sheet, respectively [20]. The ratio of the intensity of the D and G bands (I$_D$/I$_G$), can be used as an important index of the lattice defect density in the carbon structure, having a positive correlation between the I$_D$/I$_G$ ratio and the degree of defects [43]. The Raman spectra show that the I$_D$/I$_G$ ratio obviously improves from 0.86 of IGGO to 1.03 of rGOA-1. High-defect populations are found in rGOA-1 flakes, which could be attributed to the strong reducing activity of VC, which results in the oxygen-containing functional groups also being removed, leaving vacancies in the original positions. The 2D bands located in the range of 2300~3100 cm$^{-1}$ are also a characteristic feature of carbon materials [21]. In the Raman spectra (Figure 4b), rGOA-0 and rGOA-1 demonstrate a wide and up-shifted 2D peaks, indicating a structure with few layers [44].

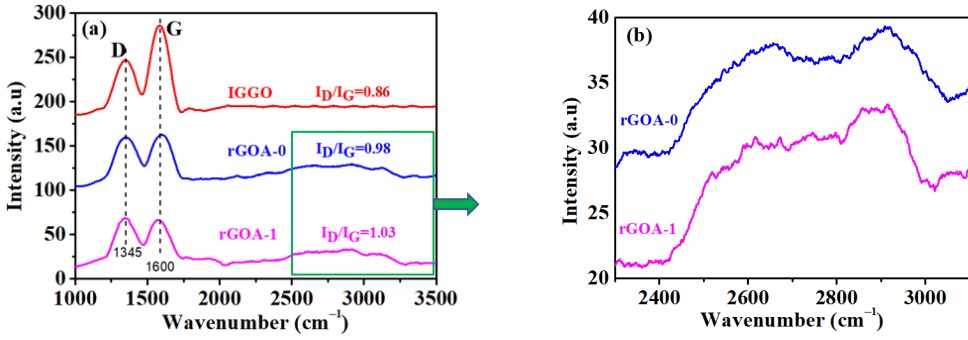

**Figure 4.** Raman spectra of IGGO, rGOA-0, and rGOA-1 (**a**) and up-shift 2D peaks in Roman spectra of rGOA-0, and rGOA-1(**b**).

FTIR spectra of IGGO, rGOA-0, and rGOA-1 (Figure 5) exhibit characteristic absorption bands. The absorption band at about 3441 cm$^{-1}$ is attributed to the O–H stretching vibration of in the carboxyl groups, C–OH groups, and adsorbed-state water. It suggested that the carboxyl group was reduced to the hydroxyl group, possibly indicating an incomplete reduction of the IGGO. Several distinct peaks of IGGO are observed, such as C–H (at 2927 cm$^{-1}$, low-intensity stretching and bending vibrations), C=O (at 1727 cm$^{-1}$, stretching vibration of carboxyl groups), C=C (at 1600 cm$^{-1}$ and 1380 cm$^{-1}$, vibration of the graphene skeleton), and C–O–C (at 1050 cm$^{-1}$, stretching vibration of epoxy groups) [45,46]. After the self-hydrothermal reduction of rGOA-0, the intensities of the peaks at 1727 cm$^{-1}$ (C=O stretching vibration) and 1050 cm$^{-1}$ (C–O–C stretching vibration) are weakened to a certain extent, illustrating that incomplete reduction takes place. Furthermore, for rGOA-1, the C=O peak at 1727 cm$^{-1}$ is absent, and the peak at 1050 cm$^{-1}$ (C–O–C stretching vibra-

tion) become very weak, indicating that the carbonyl group is completely reduced, and nucleophilic substitution occurs between the epoxy group and VC [43].

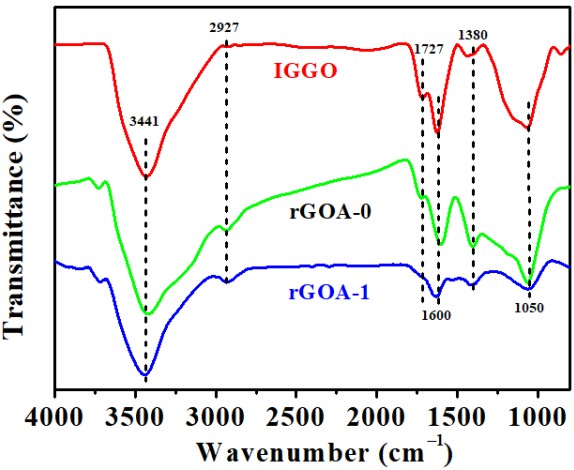

**Figure 5.** Fourier-transform infrared (FTIR) spectra of IGGO, rGOA-0, and rGOA-1.

According to the experimental results described above, the possible working mechanism is suggested, as shown in Figure 6. Under hydrothermal conditions, the carboxyl group on IGGO is reduced to the hydroxyl group by VC, and the epoxy bond dehydrates the water molecule by nucleophilic ring-opening, restoring the C=C conjugate system. Graphene sheets interweave with each other through hydrogen bonding and π–π stacking to form a three-dimensional network [47,48].

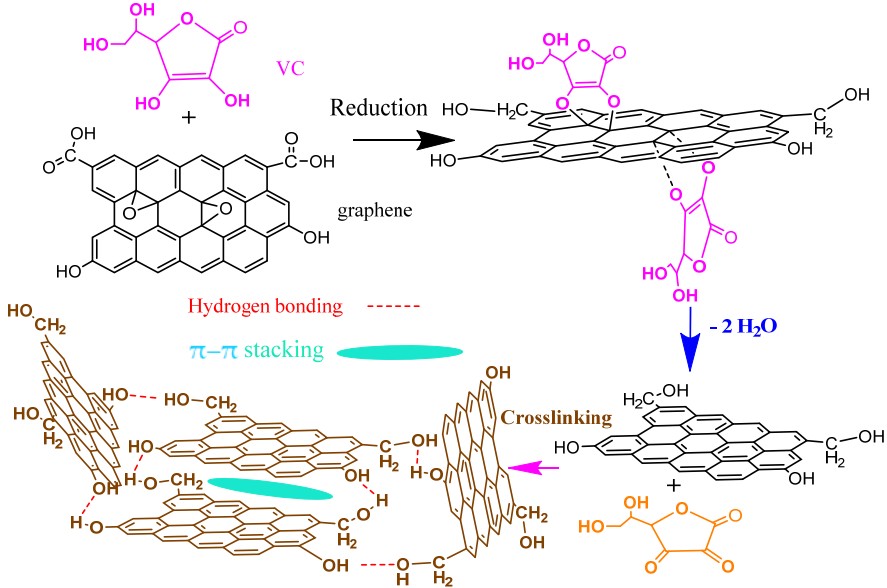

**Figure 6.** Working mechanism of VC reduction and crosslinking.

### 3.3. Comparison of Dynamic Adsorption Performances of rGOAs

The adsorption breakthrough curves of L2 were measured with IGGO and rGOAs at 20 °C, $C_{in}$ of 14.62 mg L$^{-1}$, and $V_g$ of 50 mL min$^{-1}$. As shown in Figure 7, the experimental scatterplots are fitted with the curves of the Yoon–Nelson model, and model parameters and the results of experiment are presented in Table 2. The Yoon–Nelson model showed a high relative correlation coefficient values ($R^2$) greater than 0.99, indicating that the model is right for describing the dynamic adsorption behavior of L2 on the rGOAs. Therefore, the calculated values of performance parameters ($t_{B,th}$, $Q_{B,th}$) are similar to those ($t_B$,

$Q_B$) determined experimentally. In the results that follow, $t_{B,th}$ and $Q_{B,th}$ are employed to analyze and compare the adsorption capacity of the rGOAs materials. The IGGO and rGOA-0 adsorbent beds are instantly penetrated, indicating that IGGO and rGOA-0 materials have very small L2 adsorption ability. Conversely, the breakthrough curves of other rGOAs have a longer delayed breakthrough point, illustrating a strong adsorption ability. Among the three rGOAs adsorbents, the order by adsorption ability is rGOA-1 > rGOA-0.5 > rGOA-2. rGOA-1 shows the highest adsorption capacity and the longest breakthrough time, with $t_{B,th}$ and $Q_{B,th}$ values of 9.66 min and 77.7 mg g$^{-1}$, respectively. Accordingly, $Q_{B,th}$ is expected to be roughly proportional to $S_{BET}$, $V_{tot}$, and contact angle (Figure S4), which can be hypothesized that the adsorption mechanism of rGOAs on L2 may depend on hydrophobic effect and capillary condensation [21,30].

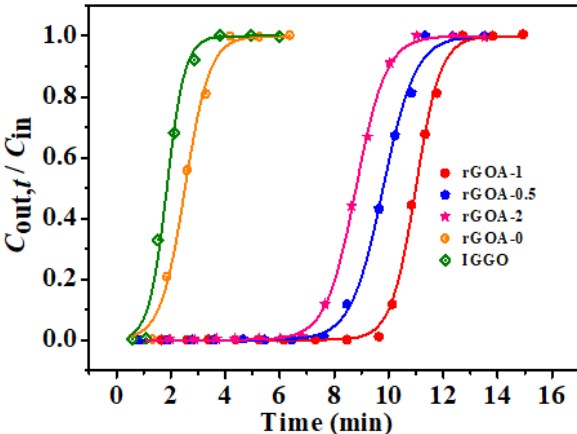

**Figure 7.** Breakthrough curves of different adsorbents for L2.

**Table 2.** Adsorption parameters of different adsorbents for L2 [a].

| | Experimental | | Model | | | | |
|---|---|---|---|---|---|---|---|
| Absorbent | $t_B$ (min) | $Q_B$ (mg g$^{-1}$) | $t_{B,th}$ (min) | $Q_{B,th}$ (mg g$^{-1}$) | $K_{YN}$ | $\tau$ (min) | $R^2$ |
| IGGO | 1.16 | 8.1 | 0.89 | 7.1 | 3.0263 | 1.86 | 0.9901 |
| rGOA-0 | 1.45 | 10.6 | 1.20 | 9.5 | 2.2160 | 2.53 | 0.9942 |
| rGOA-0.5 | 7.96 | 63.6 | 8.06 | 64.8 | 1.6900 | 9.80 | 0.9969 |
| rGOA-1 | 9.82 | 80.3 | 9.66 | 77.7 | 2.2066 | 10.99 | 0.9985 |
| rGOA-2 | 7.18 | 57.9 | 7.17 | 57.6 | 1.8042 | 8.80 | 0.9995 |

[a] $m \approx 0.10$ g, $C_{in} = 14.62$ mg L$^{-1}$, $V_g = 50$ mL min$^{-1}$.

### 3.4. Influence of Process Conditions on Adsorption Performance of rGOA-1

Inlet concentration and bed temperature are two important process parameters affecting adsorption in industrial applications [49]. Figure 8 demonstrates the parameter $t_{B,th}$ as a function of $C_{in}$. The curve exhibits an exponentially descending tendency, illustrating that a low inlet concentration is beneficial to the L2 adsorption. It is likely that these could be owing to the increased adsorption rate at higher initial concentration. Empirical Equation (3) obtained by regression analysis of experimental data can describe the trend of the curve, with the correlation coefficient values ($R^2$) greater than 0.999. This provides a theoretical reference for industrial applications of L2 adsorption.

$$t_{B,th} = 20.942 \times e^{\left(-\frac{C_{in}}{15.725}\right)} + 1.378 \tag{3}$$

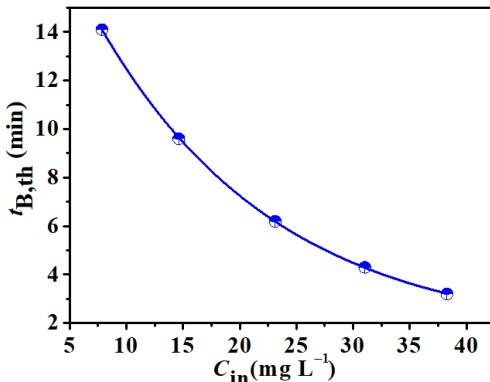

**Figure 8.** Effect of inlet concentration on breakthrough time.

Figure 9 shows the adsorption breakthrough curves for rGOA-1 at different bed temperature (0–35 °C) by the Yoon–Nelson model, with $m$ of 0.10 g, $C_{in}$ of 14.62 mg L$^{-1}$, and $V_g$ of 50 mL min$^{-1}$, respectively. In addition, the calculated parameters ($K_{YN}$, $\tau$) by the Yoon–Nelson model $t_{B,th}$ and $Q_{B,th}$ are listed in Table 3. The bed temperature decreases while the breakthrough time and adsorption capacity increase, which could be explained by the hydrophobic interaction between the rGOA-1 surface and L2 molecules and the exothermic micropore-filling adsorption of the rGOA-1 for L2 adsorbate [21,49].

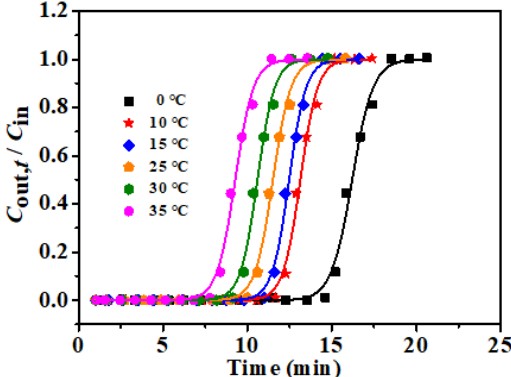

**Figure 9.** Adsorption breakthrough curves for rGOA-1 at different temperature by Yoon–Nelson model.

**Table 3.** Influence of the temperature on the adsorption of L2 over rGOA-1 [a].

| Temp. (°C) | $t_{B,th}$ (min) | $Q_{B,th}$ (mg g$^{-1}$) | $K_{YN}$ | $\tau$ min | $R^2$ |
|---|---|---|---|---|---|
| 0 | 14.38 | 104.9 | 1.6164 | 16.20 | 0.9942 |
| 10 | 11.62 | 89.2 | 1.9491 | 13.13 | 0.9956 |
| 15 | 10.99 | 85.2 | 1.9861 | 12.47 | 0.9974 |
| 25 | 9.94 | 76.3 | 1.8750 | 11.51 | 0.9964 |
| 30 | 9.11 | 69.2 | 1.9730 | 10.60 | 0.9957 |
| 35 | 7.67 | 62.1 | 1.8436 | 9.27 | 0.9951 |

[a] $m = 0.10$ g, $C_{in} = 14.62$ mg L$^{-1}$, $V_g = 50$ mL min$^{-1}$.

### 3.5. Influence of Water on the rGOA-1 Adsorbent

It is interesting to evaluate the effect of moisture on the ability of rGOA-1 to adsorb siloxanes, because biogas contains a certain amount of water [36]. As shown in Figure 10, the ability of rGOA-1 to remove L2 from dry and moist gases (RH of 70%) is similar, demonstrating that moisture does not influence the L2 removal by rGOA-1. The result is possibly due to the preceding removal of oxygen-containing groups and cross-linking of graphene layers through hydrogen bonding, providing many hydrophobic sites and leading to no polymerization reaction in the presence of moisture. As evident from Table S1,

the rGOA-1 is favorable over other kinds of adsorbents in dealing with biogas in the presence of water. Therefore, rGOA-1 can be used to remove siloxanes from moist biogases.

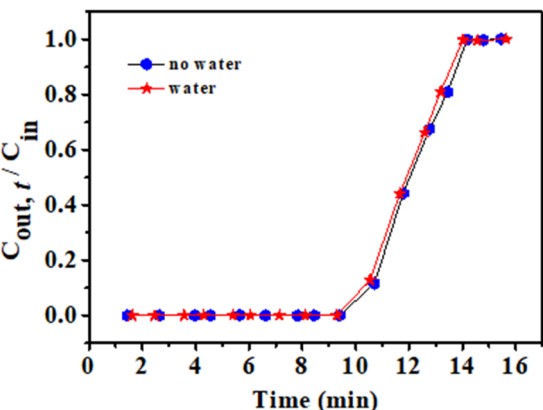

**Figure 10.** Influence of the presence of water on the adsorption of L2.

### 3.6. Recycling Performance of rGOA-1

The used rGOA-1 was recycled five times after regeneration by annealing at 80 °C for 30 min. The experimental results are presented in Figure 11, indicating that rGOA-1 possesses good regeneration properties. From Table S2, it can be observed that compared with other carbon materials, the regeneration of rGOA-1 can be achieved at a lower heating temperature, and the recovery efficiency can reach 99% without other auxiliary reagents. Although the adsorption capacity of rGOA was not high, the good regeneration performance determines its industrial application prospects.

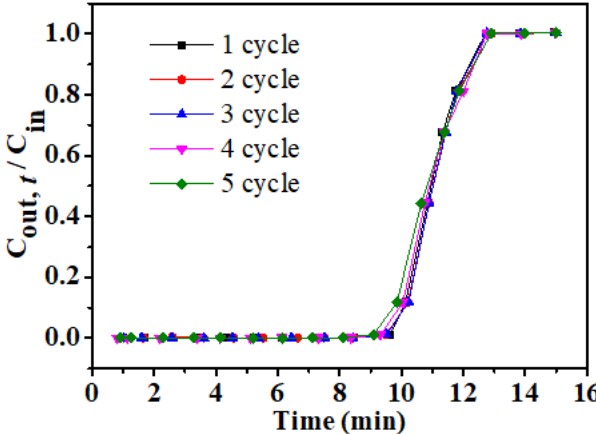

**Figure 11.** Adsorption breakthrough curves for dry L2 of rGOA-1 in the repeated cycles.

### 4. Conclusions

In conclusion, hierarchically porous 3D rGOAs could be produced using a simple one-step hydrothermal technique with VC as a reductant that assists the synthesis process. The prepared rGOA-1 exhibited excellent properties, having the highest surface area (137.9 m$^2$ g$^{-1}$) and a superior water contact angle (143.8°). Furthermore, rGOA-1 also showed the highest theoretical adsorption capacity for hexamethyldisiloxane (104.9 mg g$^{-1}$) at 0 °C. Moreover, low inlet concentration and bed temperature yielded better L2 adsorption by rGOA-1. In addition, used rGOA-1 was less sensitive to the water presence, and it exhibited good regeneration ability for multiple uses upon annealing at 80 °C. Therefore, the superior performance of rGOAs suggested its great industrial application potential.



**Supplementary Materials:** The following are available online at https://www.mdpi.com/article/10.3390/app11188486/s1, Figure S1. Molecular structure of siloxanes. Figure S2: Photographs of of the rGOAs (a: rGOA-0, b: GOA-0.5, c: rGOA-1, d: rGOA-2). Figure S3: Image showing a water droplet on the surface of IGGO and rGOAs film., Figure S4: Relationship between $Q_{B,th}$ and $S_{BET}$ (a), $Q_{B,th}$ and $V_{tot}$ (b), $Q_{B,th}$ and contact angle (c) for the five adsorbents. Table S1: Influence of the presence of water on different adsorbents for siloxanes. Table S2: Adsorption and regeneration capacities of different carbon materials for siloxanes.

**Author Contributions:** Conceptualization, Z.M. and X.M.; methodology, Z.M. and X.M.; formal analysis: Y.Z.; investigation: Y.Z., X.H. and Z.H.; resources, Z.M.; data curation: Y.Z., X.H. and Z.H.; writing—original draft preparation: Y.Z., and X.H.; writing—review and editing, Z.M. and X.M.; supervision, Z.M. and X.M.; project administration: Z.M.; funding acquisition, Z.M. All authors have read and agreed to the published version of the manuscript.

**Funding:** This research was funded by the Natural Science Foundation of Hebei Province, grant number B2021205022.

**Institutional Review Board Statement:** Not applicable.

**Informed Consent Statement:** Not applicable.

**Data Availability Statement:** Data is contained within the article or Supplementary Materials. The data presented in this study are available in Supplementary Materials.

**Acknowledgments:** The authors would like to express their gratitude to the Natural Science Foundation of Hebei Province for the financial support.

**Conflicts of Interest:** The authors declare no conflict of interest.

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
