# Peer review of "Vitamin C-Assisted Fabrication of Aerogels from Industrial Graphene Oxide for Gaseous Hexamethyldisiloxane Adsorption"

_applsci, doi:10.3390/app11188486_

Round 1

Reviewer 1 Report

The paper presents the preparation and characterization of graphene aerogels prepared via the hydrothermal method, using vitamin C as a reductant agent. The adsorption of a model molecule tests the potential application as siloxanes adsorbents from biogas.

The work is well designed and presented, nevertheless, some points should be addressed to improve it:

  1. During the entire characterization section (3.1 and 3.2), the samples rGOA-0.5 and rGOA-2 are not considered. This choice should be at least addressed at the section's beginning to help the reader. If the data are readily available should be reported in the supporting information.
  2. In general, the authors should define how they prepared the aerogels for the various characterization techniques. Indeed, during the hydrothermal reduction, a shell is often formed with properties quite different from those of the bulk (see Hu et al., doi 10.1021/acs.chemmater.5b04713, and Hu et al., doi 10.1021/acs.langmuir.5b00508). This can affect in particular SEM images and contact angle measurements (in the latter case the shell is more hydrophobic than the bulk). 
  3. Regarding the XRD patterns, the rGOA-1 pattern's intensity is very low, therefore terms like dominant (page 5, line 188) and well-ordered (page 5, line 189) should be avoided.
  4. Figure 4 is too small, it should be enlarged and, to better support the claims in the text (page 6, lines 214-215), divided into two images with a close-up of the areas of interest. Also, vertical lines identifying the peaks should be added.
  5. The infrared spectra (figure 5) show a huge OH peak, that should be absent in graphene aerogels, possibly indicating an incomplete reduction of the GO. This should be addressed in the text.
  6. The authors should clarify if the recycling experiments have been performed using dry or moist gas. Also, since one of the major drawbacks of other systems is the limited thermal regeneration in the presence of moisture the claim reported on page 10, lines 296-297, is subordinated to the recycling tests performed with the moist gas.
  7. A comparison of key performance parameters, such as tB and QB with other kinds of adsorbents would be beneficial to support the superior performance. Although the results show the L2 adsorption on the prepared aerogels, using them in industrial applications like claimed can be quite challenging. The values of tB and QB (around 10 minutes and some tenths of mg g-1) seem quite low to justify the industrial application.

Finally, some typos that should be corrected:

  1. Page 2, line 60. Authors should consistently use the nomenclature or the formulas.
  2. Page 2, line 74. Although is defined in the abstract, upon the first appearance in the main text L2 should be defined again.
  3. Page 2, line 83. "achieved"
  4. Page 3, line 95. "Brunauer–Emmett–Teller" 
  5. Page 3, line 98. "Fourier-transform infrared spectroscopy"
  6. Table 1, line 168. "Contact angle"
  7. Figure 2, line 199. add (e,f) after rGOA-1
  8. Table S1. No influence

Reviewer 2 Report

Authors have prepared the reduced graphene oxide (rGO) aerogels through hydrothermal method using vitamin C as a linker and used it for the adsorption of volatile hexamethyldisiloxane (VMS) from biogas.

It would be better to elaborate why the adsorption of VMS compounds is significant in the introduction section.

In materials and methods, outlet concentration (Cout, t) determination/measurement need to be described.

Working mechanism needs to be described to be easier to follow compared to understand from the figure.

From the FT-IR hydroxyl functional groups are present in rGOAs, which could lead to polymerization reaction in presence of moisture, which is not happed for rGOA from the moisture effect studies. Authors could explain why the hydroxyl groups in rGOA does not polymerize the siloxane in presence of moisture.

Increase in inlet concentration decrease the breakthrough time, what could be reason for faster adsorption at higher initial concentration.
